# Antioxidant, Immunostimulatory, and Anticancer Properties of Hydrolyzed Wheat Bran Mediated through Macrophages Stimulation

**DOI:** 10.3390/ijms24087436

**Published:** 2023-04-18

**Authors:** Irene Tomé-Sánchez, Cristina Martínez-Villaluenga, Ana Belén Martín-Diana, Daniel Rico, Iván Jiménez-Pulido, Juana Frias, Vermont P. Dia

**Affiliations:** 1Institute of Food Science, Technology and Nutrition (ICTAN-CSIC), José Antonio Novais, 6, 28040 Madrid, Spain; i.tome@ictan.csic.es (I.T.-S.); frias@ictan.csic.es (J.F.); 2Agricultural Technological Institute of Castilla and Leon, Government of Castilla and Leon, Finca Zamadueñas, Castilla and Leon, 47071 Valladolid, Spain; mardiaan@itacyl.es (A.B.M.-D.); ricbarda@itacyl.es (D.R.); jimpuliv@itacyl.es (I.J.-P.); 3Department of Food Science, The University of Tennessee Institute of Agriculture, Knoxville, TN 37996, USA

**Keywords:** hydrolyzed wheat bran, colorectal cancer, immunomodulation, anticancer activity, antioxidant activity, apoptosis, western blotting

## Abstract

Previous studies demonstrated that enzymatic hydrolysis enhances wheat bran (WB) biological properties. This study evaluated the immunostimulatory effect of a WB hydrolysate (HYD) and a mousse enriched with HYD (MH) before and after in vitro digestion on murine and human macrophages. The antiproliferative activity of the harvested macrophage supernatant on colorectal cancer (CRC) cells was also analyzed. MH showed significantly higher content than control mousse (M) in soluble poly- and oligosaccharides (OLSC), as well as total soluble phenolic compounds (TSPC). Although in vitro gastrointestinal digestion slightly reduced the TSPC bioaccessibility of MH, ferulic acid (FA) levels remained stable. HYD showed the highest antioxidant activity followed by MH, which demonstrated a greater antioxidant activity before and after digestion as compared with M. RAW264.7 and THP-1 cells released the highest amounts of pro-inflammatory cytokines after being treated with 0.5 mg/mL of digested WB samples. Treatment with digested HYD-stimulated RAW264.7 supernatant for 96 h showed the most anticancer effect, and spent medium reduced cancer cell colonies more than direct WB sample treatments. Although a lack of inner mitochondrial membrane potential alteration was found, increased Bax:Bcl-2 ratio and caspase-3 expression suggested activation of the mitochondrial apoptotic pathway when CRC cells were treated with macrophage supernatants. Intracellular reactive oxygen species (ROS) were positively correlated with the cell viability in CRC cells exposed to RAW264.7 supernatants (r = 0.78, *p* < 0.05) but was not correlated in CRC cells treated with THP-1 conditioned media. Supernatant from WB-stimulated THP-1 cells may be able to stimulate ROS production in HT-29 cells, leading to a decrease of viable cells in a time-dependent manner. Therefore, our present study revealed a novel anti-tumour mechanism of HYD through the stimulation of cytokine production in macrophages and the indirect inhibition of cell proliferation, colony formation, and activation of pro-apoptotic proteins expression in CRC cells.

## 1. Introduction

Regardless of all the advances in prevention, therapy, and patient prognosis, colorectal cancer (CRC) is still the second leading cause of cancer-related deaths (approximately 1 million deaths in 2020) and the third most usual cancer worldwide [1,2]. Sporadic cancer is the most frequent type (65–70%), followed by colitis-associated cancer, which is related to chronic inflammation (25%), and hereditary CRC syndrome, as familial adenomatous polyposis and Lynch syndrome (5–10%) [3,4]. Ageing, genetics, and living conditions are known to be the main risk factors for CRC [5]. Therefore, current strategies to decrease the incidence and mortality of CRC involve the early detection of the primary tumour and its precancerous stages, a shift in lifestyle and dietary habits by decreasing modifiable risk factors (western-style diet, obesity, low physical activity, smoking, and alcoholic beverage consumption), and increasing preventive factors, such as adaptation to the Mediterranean diet [6].

CRC begins when large intestinal epithelial cells accumulate genetic and epigenetic mutations in oncogenes and tumour suppressor genes, resulting in a benign adenoma that may progress later into a carcinoma and metastasize [1,3]. In addition to these genetic and epigenetic changes, the tumour microenvironment (TME) exerts an important role in CRC pathogenesis due to the interactions established between its cellular and non-cellular components and tumour cells [5,7]. Among cellular components, tumour-associated macrophages (TAMs) are the most abundant immune cells (nearly 30–50% of the total) [8]. It is well-known that TME has the capability of modifying TAM phenotype from the undifferentiated state (M0) to classically (M1), or alternatively (M2), activated macrophages through multiple signalling pathways [7,8]. While the phenotype M1 of TAMs potentially exerts anti-tumour activity by secreting pro-inflammatory cytokines, reactive oxygen species (ROS), and other inflammatory mediators, M2-TAMs lead the anti-inflammatory response and promote angiogenesis, immunosuppression, tumour initiation, and progression, being more abundant in the later stages of tumour development [8,9]. Furthermore, TME and TAMs not only play an important role in CRC development but also are highly related to the resistance to traditional treatments.

Wheat bran (WB), the outer layer of the wheat grain, is produced worldwide as a major by-product of the wheat milling industry, and it accounts for approximately 14–19% of the grain weight. WB contains high-value health-promoting components, including dietary fibre and phenolic compounds [9]. There is in vivo and in vitro evidence that WB polysaccharides are antitumour agents with immunomodulatory activity [10,11]. WB polysaccharides activate cells of the innate immune system (macrophages), stimulating the secretion of pro-inflammatory cytokines and the production of ROS via toll-like receptor 4-p38 MAPK [12] and NF-κB signalling pathways [13]. Thus, the antitumour activity of WB polysaccharides is known to be mediated through the improvement in the immune response. Similarly, dietary polyphenols are also immunomodulating compounds showing chemopreventive effect against cancer [14]. Particularly, ferulic acid (FA), the most abundant phenolic compound in WB, displays anti-inflammatory, antioxidant, and anticancer properties on several types of cancers, although its low solubility and bioavailability in aqueous medium hinder its widespread application [15].

Our research group recently applied hybrid-processing routes (including hydrothermal, enzymatic, and drying treatments) to develop a WB ingredient (hydrolyzed WB, HYD) with improved anti-inflammatory properties through the solubilisation of dietary fibre and FA [16,17,18]. The current study aimed to explore the potential antioxidant and anticancer properties mediated by immunostimulation of macrophages with HYD before and after its incorporation in a food prototype (mousse) (Figure 1). To achieve this objective, the incorporation of HYD in a standard recipe of a mousse was followed by a nutritional characterization. Antioxidant, immunostimulatory, and anticancer effects were explored in vitro through a preliminary simulation of physiological conditions in the gut using the INFOGEST 2.0 digestion model [19]. The stability of FA to gastrointestinal conditions was also studied. Antioxidant activity of intestinal digests from samples was performed using bio(chemical) assays, whereas to demonstrate anticancer properties mediated by immunostimulation, two experimental approaches were used. Firstly, murine and human macrophages (RAW264.7 and THP-1 cell lines, respectively) were treated with intestinal digests from HYD and mousse containing HYD (MH) to measure the production of pro-inflammatory cytokines (tumour necrosis factor (TNF)-α, interleukin (IL)-6, and IL-1β). Secondly, spent media from RAW264.7 and THP-1 cells exposed to intestinal digests were used to treat CRC HCT-116 and HT-29 cell lines, and inhibition of proliferation and colony formation were explored. Finally, to elucidate the underlying mechanism that may be responsible for the antiproliferative activity, apoptotic protein expression and intracellular ROS production of CRC cells were determined through western blotting and fluorescence techniques, respectively.

## 2. Results

### 2.1. Mousse Formulated with HYD Was Enriched in Total and Soluble Dietary Fibre

The nutritional composition of M and MH is shown in Table 1. Carbohydrates (83.5% dry weight, dw) were the main components of M, in the form of starch provided from tapioca and 9.6% dw of total dietary fibre (TDF), due mostly to the presence of insoluble dietary fibre (IDF) (8.3% dw) and a minor proportion of OLSC (1.3% dw). Small amounts of protein (<2% dw), fat, and ash (both <1% dw), as well as phytic acid (0.22% dw) were present in M. With the replacement of a portion of tapioca starch by HYD in the formulation, MH showed 1.04-, 2.48-, 1.95-, 15.58-, 1.13-, and 3.13-fold higher content of carbohydrates, dietary fibre, high-molecular-weight soluble dietary fibre (HMW-SDF), OLSC, fat, and phytic acid, respectively. In contrast, protein content was slightly lower in MH as compared with M. No significant differences were observed for ash values between M and MH.

### 2.2. Mousse Formulated with HYD Was Enriched in Free Phenolic Compounds That Showed High Stability to Simulated Gastrointestinal Conditions

TSPC and FA content in HYD, M, and MH before (U) and after (D) in vitro digestion are shown in Table 2. As an ingredient rich in phenolic compounds, U-HYD showed the highest concentration of TSPC and FA (predominantly occurring in the *trans*-configuration) as compared with mousses (U-M and U-MH). The incorporation of HYD in the mousse formula produced a significant increase in TSPC (U-M vs. U-MH, *p* ≤ 0.05) and FA (not detected in U-M). At the end of digestion in the upper small intestine, slightly decreased TSPC levels were observed in intestinal digests from mousses (D-M and D-MH, *p* ≤ 0.05). However, FA content remained stable at the end of digestion in D-MH despite the slight reduction observed for *trans*-FA content in intestinal digests of HYD (D-HYD) as compared with the undigested counterpart (*p* ≤ 0.05).

### 2.3. Mousse Formulated with HYD Showed Improved Antioxidant Activity Up to the End of In Vitro Digestion in the Upper Small Intestine

WB phenolic compounds are widely recognized scientifically for having high antioxidant activity due to their redox properties, playing an important role in absorbing and neutralizing free radicals [20]. As variations in the TSPC and FA were evidenced upon digestion (Table 2), the antioxidant activity of HYD, M, and MH was evaluated before and after in vitro digestion using four assays (DPPH, ABTS^•+^, ORAC, and FRAP), which included hydrogen atom transfer, single electron transfer, and/or mixed mechanisms, as well as a metal–ion chelation mechanism (Table 3). Among undigested samples, U-HYD showed the highest antioxidant activity, regardless of the antioxidant method used, consistent with its TPSC and FA content (Table 2). The addition of HYD to mousse formulation increased DPPH, ABTS radical scavenging activity, ORAC, and FRAP values (U-M vs. U-MH; *p* ≤ 0.05). Moreover, the antioxidant activity of all samples increased at the end of intestinal digestion, as compared with undigested samples, and the antioxidant activity found in D-MH was noticeably higher than that observed for D-M (*p* ≤ 0.05).

### 2.4. Intestinal Digests of HYD and MH Showed Immunostimulatory Effects on RAW264.7 and THP-1 Cells

The viability of cell treatments with intestinal digests from each sample at 0.5 mg/mL was evaluated. After the exposure of macrophages to intestinal digests from the three samples studied, no cytotoxic effect was found (viability was not affected in any experimental group as compared with untreated cells, Appendix A).

In a first attempt to determine whether HYD, M, and MH had immunostimulatory properties, cytokine production (TNF-α and IL-6) was measured upon treatment with undigested and digested samples at 0.5 mg/mL and compared with lipopolysaccharide from *Escherichia coli* O55:B5 (LPS) stimulation of RAW264.7 cells (positive control, Figure 2A,B, respectively). As indicated in Figure 2A,B, U-HYD showed the highest stimulating effect in RAW264.7 macrophages, followed by U-MH and U-M. The immunostimulatory effect of HYD remained upon digestion (U-HYD = D-HYD; *p* > 0.05), whereas a greater response was observed for D-MH as compared with U-MH (*p* ≤ 0.05). Regarding the effect of WB samples on THP-1 cytokine production (Figure 2C–E), U-HYD and U-MH showed a similar capability of increasing IL-6 levels (Figure 2D), but U-HYD stimulated more efficiently the production of TNF-α and IL-1β than undigested mousses (U-MH > U-M) (Figure 2C,E, respectively). On the other hand, the production of TNF-α, IL-6, and IL-1β was enhanced after cell treatment with D-HYD. Similarly, the IL-1β level was increased by D-MH treatment (Figure 2E). As compared with the negative control, no significant differences were observed when macrophages were treated with D-M, which would indicate a lack of immunostimulatory activity.

### 2.5. Anticancer Activity of HYD and MH Is Mediated through Its Ability to Stimulate Cytokine Production in Macrophages

#### 2.5.1. Spent Media from Stimulated Macrophages with HYD and MH before and after Digestion Demonstrated Antiproliferative Activity in CRC Cells

In a first attempt, it was investigated whether HYD and MH exerted a direct antiproliferative effect on HCT-116 and HT-29 cells. CRC cells were exposed to HYD and MH, and their intestinal digests (0.5 mg/mL) for 72 and 96 h and cell viability was measured. Cell viability of CRC cells was not significantly (*p* > 0.05) affected as compared with untreated CRC cells (Appendix A); therefore, a direct antiproliferative effect of HYD and MH was not confirmed.

In a second attempt, an indirect antiproliferative effect was evaluated in CRC cells treated for 72 h and 96 h with the spent media of stimulated RAW264.7 and THP-1 cells with undigested and digested HYD and MH (0.5 mg/mL) (Figure 3A,B, respectively). Results showed a time-dependent reduction of HCT-116 viability for treatments with spent medium from RAW264.7 cells stimulated with HYD and MH before and after digestion (*p* ≤ 0.05; Figure 3A), which were similar in magnitude or even higher than positive control (C+(+)). Comparative analysis of tested samples revealed similar antiproliferative responses, although a remarkably higher inhibition of HCT-116 proliferation was observed for D-HYD (76.94% inhibition), regardless of time of treatment (Figure 3A).

HT-29 colon cancer cells were less sensitive to spent media from stimulated RAW264.7 cells. Only cell exposure to spent media from stimulated RAW264.7 cells with D-MH for 72 h showed the ability to reduce significantly the cancer cell proliferation concerning negative control (*p* ≤ 0.05; Figure 3A). On the other hand, 96 h treatment significantly reduced the proliferation of HT-29 cells when treated with RAW264.7 cells spent medium exposed to U-MH and D-HYD and D-MH by 24.9%, 12.1%, and 43.56%, respectively, as compared with untreated cells (C−(−)) (*p* ≤ 0.05; Figure 3A).

The indirect antiproliferative effect was also evaluated using spent media from stimulated THP-1 cells (Figure 3B). As compared with untreated cells (C−(−)), there was a significant reduction (*p* ≤ 0.05) in HCT-116 viability after treatment for 72 h with spent media from THP-1 cells stimulated with LPS (C+(+)), HYD, and MH, although a time-dependent effect was clearly observed (Figure 3B). However, a similar antiproliferative response was appreciated for spent media from stimulated THP-1 cells with undigested and digested HYD and MH, regardless of treatment duration (*p* > 0.05; Figure 3B).

Experiments performed in HT-29 cells did not show antiproliferative effects for treatments with spent medium from stimulated THP-1 with undigested and digested HYD and MH for 72 and 96 h as compared with untreated cells (C−(−)) (Figure 3B), suggesting a higher resistance of this cell line to cell growth arrest or cell death.

#### 2.5.2. HYD and MH after Digestion Inhibited Colony Formation on CRC Cells

Determination of the direct effect of HYD and MH treatment on the ability to reduce the capacity of colon cancer cells to form colonies was measured (Figure 3C). HCT-116 cells treatment with LPS (C+(−)) and U-HYD showed no inhibitory effect on colony formation, whereas undigested MH reduced up to 23.4% the colony formation and intestinal digests of HYD and MH at 0.5 mg/mL and led to a greater than 50% reduction of colonies formation in HCT-116 (57.1 and 51.3%, respectively) as compared with untreated cells (C−(−)). Regarding HT-29 cells, LPS (C+(−)) and U-MH exhibited a similar ability to inhibit colony formation with respect to the negative control (*p* ≤ 0.05; Figure 3C). A point worth noting is that intestinal digests from HYD and MH were the most effective treatments to decrease the number of colonies by 44.4 and 54.7%, respectively (*p* ≤ 0.05; Figure 3C), with no significant differences between them (*p* > 0.05; Figure 3C).

#### 2.5.3. Spent Media from Stimulated Macrophages with HYD and MH before and after Digestion Inhibited Colony Formation on CRC Cells

The indirect reduction of the relative percentage of colonies formed from HCT-116 and HT-29 cells was determined using spent media from RAW264.7 cells stimulated with LPS, HYD, and MH (Figure 3D). A significant reduction of the relative percentage of colony formation was observed for HCT-116 cells exposed to growth media with LPS (C+(−)) and growth media from RAW264.7 cells incubated without and with LPS (C−(+)) and (C+(+)), respectively (*p* ≤ 0.05; Figure 3D). Furthermore, U-HYD and D-HYD significantly reduced the relative percentage of counted colonies (by 72.74 and 86.49%, respectively), while no colonies were observed after treatments with spent media from RAW264.7 cells stimulated by U-MH and D-MH (100% inhibition). These results suggest that the immunostimulatory effect of HYD and MH might be responsible for the inhibition of colony formation. Similar results were observed in HT-29, in which a significant reduction in the relative percentage of colony formation was found for D-HYD, U-MH, and D-MH as compared with untreated cells (C−(−)) (*p* ≤ 0.05; Figure 3D).

When HCT-116 cells were treated with growth media from untreated and LPS-stimulated THP-1 macrophages, a significant reduction of HCT-116 colonies was observed (by 52.9 and 76.5%, respectively) as compared with both negative and positive controls (C−(−) and C+(−), respectively) (Figure 3E). Inhibitory effects on colony formation were even higher when HCT-116 cells were incubated with spent media from THP-1 activated by U-HYD (96.3% inhibition) and D-HYD, U-MH, and D-MH (100% inhibition for each of them). Regarding HT-29 colony formation, the immunostimulatory effect of HYD and MH in THP-1 cells was linked to a significant reduction in the number of colonies, where supernatant from THP-1 exposed to the intestinal digests was the most effective treatment (*p* ≤ 0.05; Figure 3E).

### 2.6. Elucidation of the Anticancer Mechanism

#### 2.6.1. Expression of Apoptosis-Associated Proteins in CRC Cells

To decipher the molecular mechanism of the observed antiproliferative effect in colon cancer cells, levels of apoptotic proteins were analyzed. As compared with untreated HCT-116 cells, there was a significant increment of anti-apoptotic Bcl-2 protein expression in CRC cells treated with spent media from unstimulated-RAW264.7 cells with a parallel upregulation of Bax expression (1.03- and 1.42-fold increase, respectively) (Figure 4A). However, a significant downregulation of Bcl-2 expression was determined in HCT-116 cells exposed to both positive controls (C+(−) and C+(+)) as well as undigested HYD and MH, with the subsequent upregulation of pro-apoptotic Bax expression, resulting in an increased Bax:Bcl-2 ratio (1.65, 1.63, 3.82, and 11.78, respectively). The highest ratio of Bax:Bcl-2 was determined when HCT-116 cells were exposed to supernatant from RAW264.7 cells activated by the intestinal digest of HYD (Bax:Bcl-2 ratio of 12.56). On the contrary, the expression of Bax was not increased, but Bcl-2 expression was significantly downregulated by supernatant from RAW264.7 stimulated with D-MH by 10-fold, leading to an increase of Bax:Bcl-2 ratio of 9.70 in HCT-116 cells (Figure 4A). Among all the treatments, HCT-116 cells exposed to C−(+) showed the highest expression of caspase-3, and only supernatants from MH-stimulated murine macrophages increased its expression (D-MH > U-MH; Figure 4A). Regarding apoptotic protein expressions in HT-29 cells, Bax expression was significantly upregulated when HT-29 cells were treated with LPS (C+(−)), but this increment was not statistically different after their exposure to spent media from HYD-stimulated RAW264.7 cells (HYD before and after digestion). Conversely, the expression of Bcl-2 was significantly downregulated in those HT-29 cells exposed to the studied treatments. As a result, increased Bax:Bcl-2 ratios were observed for C+(−), C+(+), U-HYD, D-HYD, U-MH, and D-MH (1.56, 1.41, 4.63, 4.46, 2.45, and 7.26, respectively), whereas for C−(+), the Bax:Bcl-2 ratio was rather low (0.46). Similar to HCT-116 cells, the highest caspase-3 expression was determined in HT-29 cells treated with C−(+) as compared with untreated cells (*p* ≤ 0.05). Interestingly, a higher expression of caspase-3 protein was observed when HT-29 cells were exposed to growth media from RAW264.7 cells stimulated with undigested HYD and MH as compared with digested counterparts. As shown in Figure 4C, while C−(+) slightly increased Bax and decreased caspase-3 expressions (*p* ≤ 0.05), the expression of both pro-apoptotic proteins was highly upregulated when HCT-116 cells were exposed to supernatant from THP-1 cells stimulated with positive controls and studied samples (*p* ≤ 0.05) with respect to the negative control. Supernatants from THP-1 cells stimulated with intestinal digests showed higher Bax and caspase-3 protein expressions than those before digestion, except for MH as regards to Bax expression. A similar behaviour, with a lower intensity, was detected in Bax and caspase-3 protein expressions when HT-29 cells were treated with spent media from controls as compared with HCT-116 (Figure 4D). As previously described, supernatant from THP-1 cells treated with digested samples led to a greater expression of both pro-apoptotic proteins than those with undigested samples, apart from MH for Bax expression.

#### 2.6.2. Effect of Stimulated Macrophage Spent Medium on Mitochondrial Depolarization

Mitochondrial membrane depolarization could be used to determine a loss of mitochondrial health by a decrease in the red/green fluorescence intensity ratio. According to our results, there was no loss of the mitochondrial potential by CRC cells after the treatment with spent medium from stimulated RAW264.7 and THP-1 cells as compared with the negative control (Appendix A).

#### 2.6.3. Effect of RAW264.7 and THP-1 Cells Spent Medium on Intracellular ROS Production

Reactive oxygen species (ROS) are intermediates of cellular metabolism and play an essential role in cell signalling transduction and homeostasis [21]. The variation in the intracellular ROS levels in HCT-116 and HT-29 cells after spent medium treatment was measured using 2′,7′-dichlorohydrofluorescein diacetate (H_2_DCFDA) as a fluorescent probe. As shown in Figure 5A, a significantly lower level of intracellular ROS was observed when HCT-116 cells were treated for 72 h with RAW264.7 spent medium as compared with untreated HCT-116 cells. However, HCT-116 cells exposed to supernatant from RAW264.7 stimulated with HYD and MH before and after digestion showed the lowest accumulation of intracellular ROS, and similar results were observed after 96 h of treatment (Figure 5A). Intracellular ROS levels were also measured in HT-29 cells after 72 h of treatment where no significant differences were found using negative and positive control treatments (C−(−) and C+(−)). As compared to C−(−), HT-29 cells showed the lowest level of intracellular ROS when exposed to harvested media from RAW264.7 stimulated with C+(+), HYD, and MH (before and after digestion) (*p* ≤ 0.05, Figure 5A). Interestingly, longer treatment periods (96 h) resulted in higher intracellular ROS production (*p* > 0.05). After 72 h, there was not a significant reduction of intracellular ROS when HCT-116 cells were treated with spent medium from THP-1 cells stimulated with analysed samples before and after digestion (Figure 5B). Nevertheless, 96 h treatment with undigested samples and D-MH led to a significantly lower intracellular ROS production relative to negative control (C(−)). While similar results were observed after 72 and 96 h of treatment in HT-29 cells exposed to THP-1 cells spent medium, a significantly lower level of intracellular ROS was measured in those HT-29 cells treated with both MH-stimulated THP-1 supernatants for 72 h as compared with the negative control (C−(−)) (Figure 5B).

## 3. Discussion

Optimized enzymatic hydrolysis has been demonstrated to be a sustainable and technologically efficient strategy for WB conversion into a functional ingredient with an improved nutritional profile and bioactivity [18]. In particular, HYD produced by enzyme treatment (Ultraflo XL), centrifugation, and spray-drying of the soluble fraction was composed mainly of carbohydrates including mono- and disaccharides (20% dw), starch (10% dw), and SDF (32% dw). The latter was characterized by the presence of β-glucan (2.5% dw) and OLSC (22% dw), where xylotriose, 3^3^-α-L-arabinofuranosyl-xylotetraose/3^3^-α-L-plus-2^3^-α-L-arabinofuranosyl-xylotetraose, 2^3^,3^3^-di-α-L-arabinofuranosyl-xylotriose, and O-(5-O-feruloyl-α-L-arabinofuranosyl)-(1-3)-O-β-D-xylopyranosyl-(1-4)-D-xylopyranose were identified. In addition, HYD is a source of free phenolic compounds (0.93 g GAE/100 g dw), of which 89% was as FA [18].

The first aim of this work was to evaluate the effect of HYD incorporation into a mousse formulation. Developed MH (containing 44% of HYD, dw) showed a remarkable enrichment in HMW-SDF, OLSC, and TSPC, consistent with the nutritional composition of HYD (Table 1). Unlike dietary fibre compounds, the stability and bioaccessibility of phenolic compounds in the gastrointestinal tract may vary as a function of the food matrix, food processing, and physiological conditions during food digestion [22]. The lack of stability in the gastrointestinal tract has been supported by many investigations to reduce the bioaccessibility of phenolic compounds [22]. Thus, to attribute biological activity to phenolic compounds in HYD and MH, it was critical to understand their digestive stability and bioaccessibility. In the present study, TSPC in HYD showed an acceptable stability to simulated gastrointestinal conditions. Hydrophilic phenolic acids in HYD have been reported to be stable to gastroduodenal digestion [18]. In addition, FA in HYD was slightly reduced at the end of digestion, likely due to its chemical instability at the near alkaline pH of pancreatic fluids [23]. The chemical nature of the food matrix can also affect the release of dietary polyphenols [22]. The MH food matrix composed mainly of starch and SDF could favour the entrapment of free phenolic compounds coming from HYD during digestion. This fact could be a plausible explanation for the lower content of TSPC observed in MH after gastroduodenal digestion. Polyphenols may be bound to food components, such as dietary fibre or resistant proteins, reducing their bioaccessibility in gastric and pancreatic juices [22].

The second aim of this work was to evaluate the antioxidant, immunostimulatory and anticancer properties of HYD and MH. Regarding antioxidant activity, HYD was confirmed to inhibit free radical scavenging through distinct mechanisms of action, including hydrogen atom donation, electron donation, and ion–metal chelation (Table 3). The incorporation of HYD into the mousse formulation greatly improved the antioxidant activity, which could be attributed to the phenolic compounds. This observation was supported by a positive correlation between TSPC and antioxidant assays (r = 0.94, *p* < 0.05 for DPPH and r = 0.92, *p* < 0.05 for ORAC), in line with similar findings reported by other authors [24,25]. As previously described, the release of other antioxidant compounds due to the enzymatic hydrolysis and the increase of environmental pH during the in vitro digestion, which leads to the deprotonation of hydroxyl groups from phenolic aromatic rings, may be responsible for the increased antioxidant effects of HYD and MH, among others [18].

To evaluate anticancer ability mediated by immunostimulation, two experimental approaches were used. Firstly, due to their different tolerances and responses against stimuli [26], two macrophage cell lines (murine RAW264.7 and human THP-1 macrophages) were treated with HYD and mousse containing HYD (MH) before and after in vitro digestion to measure the production of pro-inflammatory cytokines (TNF-α, IL-6, and IL-1β). Secondly, spent media from RAW264.7 and THP-1 cells exposed to undigested samples and intestinal digests were used to treat HCT-116 and HT-29 cell lines, and inhibition of proliferation and colony formation were analysed.

Regarding cytokine production, results showed that HYD and MH at the studied dose (0.5 mg/mL) have an immunostimulatory effect on RAW264.7 and THP-1 macrophages by inducing the production of pro-inflammatory cytokines (TNF-α, IL-6, and IL-1β). These results may be attributed to the presence of HMW-SDF (9.74% for HYD and 1.68% for MH) that can enhance the expression of pro-inflammatory mediators (TNF-α, IL-6, and cycloxygenase-2) in unstimulated RAW264.7 cells [27]. Among soluble polysaccharides, WB arabinoxylans can induce the production of nitric oxide (NO) in a dose-dependent manner, as well as the production of MCP-1, TNF-α, and IL-6 on RAW264.7 macrophages via the NF-κB signalling pathway [13]. Similar results were reported when RAW264.7 cells were treated with phenolic acid bound arabinoxylans isolated from two different varieties of millets [28]. Smaller molecules, such as xylooligosaccharides, also induce the release of TNF-α, IL-1β, IL-6, and NO production in unstimulated RAW264.7 cells [29]. On the other hand, the immunomodulatory activity of soluble cereal poly- and oligosaccharides has been also described in human macrophages. β-glucan can activate THP-1 macrophages as a key pathogen-associated molecular pattern (PAMP) that triggers the host’s immune response [30]. For example, THP-1 cells treated with 100 µg/mL of β-glucan derived from oat and barley showed the ability to increase the expression of IL-1β, IL-8, and NF-ĸB genes after 3 and 6 h of stimulation [31]. Recently, a maltoheptaose OLSC from wheatgrass, with the ability to act as an immune stimulator of human THP-1 macrophages via Toll-like receptor-2 signalling, was identified [32]. The immunostimulatory response of HYD and MH remained or increased after digestion, suggesting that digestion conditions may have favoured the release of compounds with biological activities, as has been recently reported in digested bioprocessed wheat ingredients [33]. Slight differences among treatments were found when comparing the cytokine production of the two cell lines. When digested, HYD and MH enhanced the cytokine levels in human and mouse macrophages, respectively. In the same way, digested MH increased the release of IL-1β in THP-1 cells.

Previous studies reported that macrophages may carry out anticancer effects by secreting NO, intracellular ROS, and pro-inflammatory cytokines [34,35]. Thus, we studied the potential anticancer activity of the harvested stimulated macrophage supernatants containing pro-inflammatory molecules. While direct incubation of CRC cells with WB samples did not show an obvious cytotoxic effect despite digestion conditions or treatment duration, spent medium from WB-stimulated macrophages was able to modify CRC cell proliferation after 72 and 96 h. Similar results were observed when HCT-116 and RKO cells were exposed to fucoidan (FPS1M), a polysaccharide extracted from brown algae [36]. In this study, direct incubation of CRC cells with FPS1M did not show cytotoxicity, while conditioned media from RAW264.7 macrophages treated with FPS1M decreased cell viability and promoted cell apoptosis.

Interestingly, different responses to the treatments with conditioned media were observed in CRC cells. In general, a higher antiproliferative effect was observed in those CRC cells treated with spent media from stimulated RAW264.7 than from THP-1 cells. Spent media from WB-treated RAW264.7 cells exerted a more efficient antiproliferative effect on HCT-116 cells than on HT-29 cells, suggesting a higher sensitivity of HCT-116 cells to these supernatants. Similarly, in a short-term cell viability assay, a higher sensitivity was found in HCT-116 cells as compared with HT-29 after their exposure to different anticancer molecules (docetaxel, oxaliplatin, 5-fluorouracil, and camptothecin) that increased in a time-dependent manner (from 24 to 72 h). In accordance with our results, researchers concluded that this different sensitivity may be related to the parental cell characteristics rather than to the drug type used [37]. In addition, digestion conditions and longer incubation times (96 h) improved the indirect antiproliferative effect of HYD-stimulated RAW264.7 cells, resulting in these treatment conditions being more efficient at reducing cell viability. Culture supernatant from THP-1 cells stimulated with WB samples decreased HCT-116 cell viability after 72 and 96 h, while a time-response dependency was observed in HT-29 cells, as a high reduction of viable cell percentage was observed after 96 h of treatment. This result suggests that a longer treatment duration with WB sample-stimulated human macrophages may induce a delayed antiproliferative effect on HT-29 cells.

To evaluate whether studied samples might prevent the formation of CRC colonies, a colony formation assay was performed. Digestion conditions improved the direct capability of WB samples to inhibit the colony formation of CRC cells. However, better results were observed after using supernatants from immunomodulated macrophages, reducing more efficiently the percentage of colonies in HCT-116 cells than in HT-29 cells.

Finally, underlying molecular mechanisms of the observed antiproliferative effect of WB-stimulated macrophages spent medium on CRC cells were faced. Cell proliferation occurs through a sequence of harmonized changes known as the cell cycle, where apoptosis, or programmed cell death, is necessary to maintain balanced tissue homeostasis [38,39]. The loss of apoptotic processes as well as uncontrolled cell growth and differentiation lead to cancer development [40]. One of the approaches to combat cancer progression involves triggering cellular apoptosis signalling pathways in those aberrant cells [41]. Among cellular events involving apoptosis, the loss of inner mitochondrial transmembrane potential (Δψm) is one of the most indicative processes at an early stage [42]. Fluorescence data showed that collected media from WB-stimulated macrophages did not produce a significant alteration of mitochondrial membrane potential as compared with untreated cells, even following long-time exposure treatments. Because we observed a decrease in CRC cell proliferation, other biochemical parameters were evaluated to understand the cellular status and to check whether other levels were affected after treatments.

Many factors (mostly proteins) have been identified as playing a key role in apoptosis. The expressions of Bcl-2, Bax, and caspase-3 were modified in colon cancer cells treated with stimulated macrophage supernatants, which indicates an activation of the mitochondrial apoptotic pathway (Figure 4). Intrinsic or mitochondrial apoptotic pathway is dependent on proteins from the Bcl-2 family, where protein Bcl-2 is a key inhibitor of apoptosis and is usually found over-expressed in tumours, such as CRC [43]. In addition, high levels of anti-apoptotic Bcl-2 family members are associated with resistance of many tumours to chemotherapy [44]. Supernatants from murine macrophages stimulated with HYD and MH led to a significant decrease of Bcl-2 expression in colon cancer cells, resulting in a higher Bax:Bcl-2 ratio. The highest antiproliferative effect and Bax:Bcl-2 ratios were observed when HCT-116 and HT-29 cells were treated with spent media from RAW264.7 cells activated by intestinal digests of HYD and MH, respectively (Figure 3A). In cancer therapy, evidence shows that high levels of Bax:Bcl-2 ratio may result not only in decreased resistance to apoptosis but also in a better prognosis and a lower metastasis [45]. Thus, WB samples could be promising ingredients for CRC prevention. On the other hand, the pro-apoptotic protein Bax belongs to the Bcl-2 family, and its activation results in the permeabilization of the outer mitochondrial membrane, leading to the release of pro-apoptotic molecules, such as cytochrome C, which combined with caspase-9, activates caspase-3 and, finally, the execution pathway of apoptosis [46]. Spent media from WB-stimulated macrophages increased pro-apoptotic Bax expression and, hence, upregulated caspase-3 expression, which may be responsible for the anticancer effect observed.

Among other characteristics, tumour cells express lower antioxidants than normal cells, leading to a higher ROS level and, therefore, a persistent pro-oxidative state [47,48]. ROS quantification indicated that colon cancer cells treated with spent medium from WB-stimulated murine macrophages showed significantly lower values of intracellular ROS, and there was a reduction in a time-dependent manner when CRC cells were treated with WB-stimulated human macrophages (Figure 5). This finding could be related to the observed anticancer effect of WB-stimulated macrophage supernatants (Figure 3A,B) that was confirmed by a positive correlation between intracellular ROS levels and viable cells percentage (r = 0.78, *p* < 0.05).

## 4. Materials and Methods

### 4.1. Chemicals, Standards, and Reagents

Fast Blue BB (FBBB) [4-benzoylamino-2,5-dimethoxybenzenediazonium chloride hemi-(zinc chloride) salt], bile extract porcine, pancreatin from porcine pancreas, pepsin from porcine gastric mucosa, α-amylase from human saliva, ABTS, 2,2′-diazobis-(2-aminodinopropane)-dihydrochloride (AAPH) fluorescein, DPPH, 2,4,6-tripyridyl-s-triazine (TPTZ), ferric chloride hexahydrate, and phorbol 12-myristate 13-acetate (PMA) were purchased from Sigma-Aldrich Co. (St. Louis, MO, USA). Standards of Trolox, gallic acid, and FA were acquired from Sigma-Aldrich Co. (St. Louis, MO, USA), and ferrous sulfate from Panreac (Montplet & Esteban, S.L., Barcelona, Spain). Laemmli sample buffer (2×) and Clarity MaxTM Western ECL Substrate were obtained from Bio-Rad Laboratories (Hercules, CA, USA). CellTiter 96^®^ AQ_ueous_ Non-Radioactive Cell Proliferation Assay Kit was obtained from Promega (Madison, WI, USA). Radioimmunoprecipitation assay (RIPA) buffer, Halt^TM^ Protease Inhibitor Cocktail, Bradford Dye Reagent, Tris Base, and Tween^®^ 20 were obtained from Thermo Fisher Scientific (Waltham, MA, USA). JC-1 Mitochondrial Membrane Potential Dye and H_2_DCFDA were purchased from Biotium Inc. (Hayward, CA, USA). GAPDH, Bax, Bcl-2, and caspase-3 primary antibodies were acquired from Proteintech Group Inc. (Rosemont, IL, USA). Antimouse IgG horseradish peroxidase and anti-rabbit IgG alkaline phosphatase conjugate secondary antibodies were purchased from GE Healthcare (Buckinghamshire, UK).

### 4.2. Raw Material

WB (*Triticum aestivum* L. var. Craklin) (<800 µm) was obtained from a Spanish wheat flour producer, Emilio Esteban, S.A. (EMESA, Valladolid, Spain), and kept in vacuum-sealed plastic bags at room temperature (20 ± 2 °C) in dark conditions for further analysis.

### 4.3. Pilot-Scale HYD Production

The production and stabilization of HYD through enzymatic hydrolysis and spray-drying was performed at a pilot scale according to the previously described processing conditions with slight modifications [17]. First, 20 kg of WB were soaked in 200 L of sterile water (ratio 1:20, *w/v*). Enzymatic hydrolysis was conducted using a BIOSTAT F300^®^ bioreactor coupled to a dynamic control unit with 50% shaking (Sartorius-Stedim Biotech, Barcelona, Spain) by adding Ultraflo XL at 2% (enzyme to WB dry weight ratio, *w*:*w*) under the following conditions: 47 °C, pH 6, and 20 h. A final volume of 120 L of HYD with a solubilisation yield of 32.9% was obtained after centrifugation using a CLARA 20 Low Flow separator (Alfa Laval Iberia S.A., Madrid, Spain). Then, soluble fraction stabilisation was performed by spray-drying through a rotary spray dryer (Instalaciones Indrustriales Grau, S.R.L., Valencia, Spain), being fed into the main chamber using a peristaltic pump, with a flow rate maintained at 130 m^3^/h and compressor air pressure of 3 bar. The inlet/outlet drying temperatures were set at 180/85 °C. The powder was collected at the exit of a cyclone.

HYD showed the following composition: 31.65% of TDF, 9.74% of HMW-SDF, 21.91% of OLSC, 10.36% of proteins, and 0.85% of phytic acid [18].

### 4.4. Production of Mousses

The formula M was prepared by mixing 50 g of water, 11.5 g of tapioca starch, 0.5 g of cocoa, 0.1 g of cinnamon, and 5 g of sugar. In MH, 7.5 g of starch tapioca was replaced with an equal amount of HYD. The mixture was boiled at 85 °C, ensuring the consistency of heavy cream and a homogeneous thick mixture of all the ingredients. The final formula was allowed to cool down to 40 °C and 1.5 g methylcellulose was added and mixed with a food blender for complete homogenisation. The mixture was then kept at 4 °C for 6–8 h. After that, whipping of the mixture was carried out until a foamy texture was obtained. A volume of 50 mL of the mixture was placed into individual sterile recipients and baked at 66 °C for 8 min. Finally, the mousses were immediately cooled down and frozen until further use.

### 4.5. Simulated Gastrointestinal Digestion

The intestinal digests of HYD, M, and MH were obtained using the INFOGEST 2.0 method [19], as previously reported by Tomé-Sánchez et al. [18] with minor modifications. Three grams of sample were accurately weighed and, consecutively, mixed in a ratio of 1:1 with simulated saliva (*w*/*v*), gastric (*v*/*v*), and intestinal (*v*/*v*) fluids. Previously, amylase, pepsin, and pancreatin activities and (bile concentration) were determined and used in digestion assays at a final concentration in the simulated fluids of 75 U/mL, 2000 U/mL, and 800 U/mL, and 10 mmol/L, respectively. The enzymatic activity in intestinal digests was inactivated in a boiling water bath for at least 10 min. Samples were freeze-dried (Virtis Company, Inc., Gardiner, NY, USA) and stored in refrigerated conditions until further analysis. The digestion process was performed in duplicate.

### 4.6. Chemical Characterization

Moisture content was analyzed gravimetrically by drying samples at 100 °C for 24 h [49]. Fat content was measured gravimetrically after fat extraction over 4 h in a Soxhlet using petroleum ether at 40–60 °C. Nitrogen content was determined by the Dumas method (AOAC, 2005), and protein was calculated from nitrogen content using the conversion factor of 5.25. Ash content was determined following method 923.03 (AOAC, 1990). Carbohydrates were estimated by difference. Phytic acid was determined using the K-PHYT assay kit (Megazyme, Wicklow, Ireland). Total dietary fibre and its fractions were carried out using a K-RINTDF assay kit (Megazyme, Wicklow, Ireland). All the parameters were expressed as g per 100 g of sample (dw).

Total soluble phenolic compounds (TSPC) were obtained after two serial extractions with 1 mL of 0.1% of formic acid in 80% methanol followed by another extraction with 1 mL of 0.1% of formic acid in 70% acetone and analysed by FBBB reaction [50] according to Tomé-Sánchez et al. [18]. TSPC were quantified using a linear standard curve of gallic acid (0–225 µg/mL) with R^2^ > 0.99. Results were expressed as mg of GAE per 100 g of dw sample.

The identification of soluble FA isomers was performed using an Alliance Separation Module 2695 (Waters, Milford, MA, USA) coupled to a diode array detector (model 2998, Waters, Milford, MA, USA). Briefly, 20 µL of the soluble phenolic fraction was injected into a C18 Alltima analytical column (250 mm × 4.6 mm × 5 µm) equipped with a C18 Alltima precolumn (7.5 mm × 4.6 mm × 5 µm) (Grace Davison Discovery Sciences, Deerfield, IL, USA), both maintained at 30 °C. The samples were eluted for 50 min at an isocratic flow rate of 1 mL/min. The mobile phase consisted of deionized water and formic acid (99:1, *v*/*v*). Data acquisition and integration were performed using Empower II software (Waters, Milford, MA, USA). FA isomers were monitored at 280 nm and identified by retention time and spiking the sample with a standard solution. Quantification of FA isomers was determined by using a standard curve of authentic FA (0–450 µg/mL linear range, R^2^ > 0.99). Data were expressed as mg of FA per 100 g dw. All analyses were carried out in duplicate.

### 4.7. Antioxidant Activity

#### 4.7.1. Sample Preparation

One gram of undigested samples was extracted with 10 mL of acidified methanol: water (ratio 50:50, *v*/*v*; pH 2) on a temperature-controlled orbital shaker (250 rpm, 1 h, 25 °C). Samples were centrifuged (1800× *g*, 10 min, 25 °C), and the supernatant was collected and filtered using Whatman paper n.1 (Sigma-Aldrich Co., St. Louis, MO, USA). The extraction was repeated and the combined methanol fractions were adjusted to 25 mL. Then, aliquots of the extract were created and stored at −80 °C until further analysis. On the other hand, ten milligrams of digested samples were dissolved in 1 mL of water and stored at 4 °C until further analysis.

#### 4.7.2. DPPH Radical Scavenging Activity

DPPH radical scavenging capacity was analyzed by using the method described by Brand-Williams et al. [51] with slight modifications. Briefly, 0.25 mL of a mixture containing sample extract, Milli-Q water, and 120 µM DPPH methanol solution (ratio 1:4:5, *v*:*v*:*v*) was added onto a 96-well microplate. Absorbance at 525 nm was recorded after 30 min with a Spectrostar Omega microplate reader (BMG Labtech, Ortenberg, Germany). DPPH values were calculated using a standard curve of Trolox solution (7.5–210 µM) and expressed as mmol of TE per 100 g dw.

#### 4.7.3. Oxygen Radical Absorbance Capacity (ORAC) Assay

ORAC assay was based on an adaptation of the procedure previously described by Ou et al. [52]. Briefly, the fluorescence intensity of extracts, standards, and blanks diluted in phosphate buffer (75 mM, pH 7.4) was measured using a Clariostar Omega microplate reader for 2.5 h at λ_exc_ = 485 nm and λ_em_ = 520 nm. Trolox solution in different concentrations was used to obtain a standard curve (7.5–210 µM) based on the net area under the curve between the blank and the sample, and it was used to calculate ORAC values, expressed as mmol TE per 100 g dw.

#### 4.7.4. ABTS Radical Cation Scavenging Activity

The ABTS^•+^ assay was carried out according to the procedure described by Miller and Rice-Evans [53]. A stock solution of 7 mM ABTS^•+^ and 2.45 mM potassium persulfate (ratio 1:1, *v*:*v*) was prepared and kept in darkness at room temperature overnight. Then, the working solution was adjusted to an absorbance of 0.7 ± 0.02 at 730 nm using a 75 mM phosphate buffer solution (pH 7.4) and equilibrated at 30 °C. The analysis was performed by adding 10 µL of the extracts, blank or Trolox standards (7.5–210 µM) to 190 µL of the ABTS^•+^ solution. Absorbance was obtained at 730 nm using a Spectrostar Omega microplate reader. ABTS^•+^ values were expressed as mmol TE per 100 g dw.

#### 4.7.5. Ferric Reducing Antioxidant Power (FRAP)

Reducing power was determined using the methodology described by Benzie and Strain [54]. The FRAP reagent was freshly prepared by mixing 300 mM acetate buffer (pH 3.6), 10 mM TPTZ dissolved in 40 mM hydrochloric acid, and 20 mM of ferric chloride hexahydrate solution in a volume ratio of 10:1:1, respectively. A reaction mixture was obtained by adding 190 µL of FRAP reagent to 10 µL of extracts. Deionized water was used as blank and ferrous sulfate heptahydrate solution (0.4 mM–3 mM) as standard. Absorbance was measured on a Spectrostar Omega microplate reader at 593 nm. Results were expressed as mmol Fe^+2^ equivalents per 100 g dw.

### 4.8. Immunostimulatory Activity

#### 4.8.1. Cell Lines and Cell Culture

All cell lines were purchased from American Type Culture Collection (Manassas, VA, USA). Human leukemia monocyte cell line THP-1 was grown in complete Roswell Park Memorial Institute (RPMI) 1640 (containing 300 mg/mL of L-glutamine) composed of 1% of penicillin–streptomycin (P-S, Mediatech Inc., Manassas, VA, USA), 10% of heat-inactivated fetal bovine serum (FBS), and 50 µM β-mercaptoethanol purchased from Gibco (Life Technologies Corporation, Grand Island, NY, USA). Murine macrophage cell line RAW264.7 was cultured in high-glucose Dulbecco’s modified Eagle’s medium (DMEM) with L-glutamine (Mediatech Inc., Manassas, VA, USA), 10% FBS, and 1% P-S (complete DMEM).

#### 4.8.2. Cell Viability of RAW264.7 and PMA-Treated THP-1 Cells

RAW264.7 macrophages were seeded at a density of 1 × 10^4^ cells/well in 96-well plates in complete DMEM and allowed to attach overnight in humidified 5% CO_2_ incubator at 37 °C. Twenty-five thousand THP-1 cells per well were seeded in 96-well plates with complete RPMI-1640 medium and 162 nM of PMA for monocyte differentiation into macrophage-like cells. Cells were incubated at 37 °C with 5% CO_2_ for 48 h and for an additional time of 24 h, in which time the spent medium was replaced with a complete RPMI-1640 medium. RAW264.7 and THP-1 macrophages were treated with undigested and digested samples at 0.5 mg/mL of complete medium for 24 h. After incubation, the spent medium was replaced with 100 µL of serum-free medium with CellTiter 96^®^ AQueous Non-Radioactive Cell Proliferation Assay solution (ratio 9:1, *v*/*v*) for 2 h at 37 °C in a humidified incubator with 5% CO_2_. Absorbance was read at 490 nm on an EL808 microplate reader (Biotek Instruments, Winooski, VT, USA). Untreated RAW264.7 and THP-1 cells were considered as negative control. Cells stimulated with LPS (Sigma-Aldrich Co., St. Louis, MO, USA) were included as a positive control. Cell viability was expressed as a percentage of untreated cells (negative control, C−). Data represent the mean and the standard deviation of eight biological replicates.

#### 4.8.3. Cytokine Production

Cytokine concentration was determined in spent medium from treated macrophage cells in the above specified conditions. TNF-α, IL-6, and IL-1β were measured by enzyme-linked immunosorbent assay kits following the manufacturer’s instructions (BioLegend, San Diego, CA, USA). The absorbance was measured with an EL808 microplate reader at 450 nm. Results were expressed in pg/mL. Data represent the mean and standard deviation of eight biological replicates.

### 4.9. Anticancer Activity

#### 4.9.1. Cell Lines and Cell Culture

Human epithelial colon cancer cell lines (HCT-116 and HT-29) were grown in complete Stable CellTM Minimum Essential Medium (MEM) made up of Earle’s salts, glutamine, and sodium bicarbonate, 10% of FBS, 1% of P/S, 1% of MEM non-essential amino acid solution, and 1% of 100 mM sodium pyruvate (Sigma-Aldrich Co., St. Louis, MO, USA) (complete MEM).

#### 4.9.2. Antiproliferative Effect on Colorectal Cancer Cells

HCT-116 and HT-29 cells were seeded (5 × 10^3^ cells/well) in 96-well plates and grown in complete MEM at 37 °C with 5% CO_2_. The direct antiproliferative effect of undigested and digested samples was performed by cell treatment at 0.5 mg/mL in serum-free MEM for 72 and 96 h. This dose was selected on the basis of preliminary studies in which 0.5 mg/mL was an effective dose in anti-inflammatory activity assays [16,17,18]. The indirect antiproliferative effect was determined by exposing HCT-116 and HT-29 cells for 72 and 96 h to spent medium collected from RAW264.7 and THP-1 cells treated with undigested and digested samples at 0.5 mg/mL in serum DMEM or RPMI-1640, respectively. After indicated treatment duration, the medium was removed, and 100 μL free serum media and 20 µL CellTiter 96^®^ Aqueous Non-Radioactive Cell Proliferation Assay solution were added. Plates were incubated at 37 °C for 2 h, and absorbance was read at 490 nm in an EL808 microplate reader. Cell viability was expressed as percentage of untreated cells (negative control, C−(−)). The positive control was LPS-treated cells (1 µg/mL) without sample extracts (C+(+)). Data represent the mean and standard deviation of eight biological replicates.

#### 4.9.3. Colony Formation Assay

HCT-116 and HT-29 cells were seeded (2 × 10^3^ cells/well) in 24-well plates, grown in complete MEM, and allowed to attach overnight. Cells were exposed to 0.5 mg/mL of undigested and digested samples or spent medium collected from RAW264.7, and THP-1 cells were exposed to undigested and digested samples (0.5 mg/mL) for 7 days of incubation in a controlled environment (5% CO_2_ at 37 °C). Then, the medium was removed and cell colonies were fixed and stained with a methanol solution containing crystal violet stain (0.5%) for 1 h at room temperature. After removing the excess of the staining solution, plates were carefully washed with distilled water and dried at room temperature for at least 24 h. Results were expressed as inhibition of colony formation relative to the untreated cells (C−(−)) after counting cell colonies (colonies were groups of at least 50 cells). Data represent the mean and standard deviation of four biological replicates.

#### 4.9.4. Apoptosis Assay

Apoptosis was determined by fluorescence spectrophotometry according to Sivandzade et al. [55] with slight adaptations. Treated CRC cells were washed twice with warm PBS and incubated for 30 min at 37 °C in dark conditions with 150 µL of 2 µM JC-1 dye in PBS. After staining and washing with PBS, 100 µL of the sample was added in a 96-well black plate. Red fluorescence was measured at λ_exc_ = 550 nm and λ_em_ = 600 nm, and green fluorescence at λ_exc_ = 485 nm and λ_em_ = 535 nm, using a Synergy H1 Hybrid Multi-Mode Reader. Apoptosis was determined as the ratio of red fluorescence to that of green fluorescence and was expressed as the percentage relative to untreated cells (C−(−)). Data represent the mean and the standard deviation of eight biological replicates.

#### 4.9.5. Immunoblotting

Lysates of treated cells were obtained after cell washings with ice-cold PBS and incubation for 5 min with ice-cold RIPA lysis buffer containing a protease inhibitor cocktail. Then, scrapped cell lysates were collected, vortexed for 5 min, and centrifuged at 14,000× *g* for 10 min at 4 °C in a Sorvall Legend Micro 21R Centrifuge (Thermo Fisher Scientific, Waltham, MA, USA). Lysate aliquots were diluted with deionized water for soluble protein quantification by Bradford assay, using bovine serum albumin as standard. The remaining supernatants were collected and diluted in Laemmli buffer (ratio 1:1, *v*/*v*) with 5% β-mercaptoethanol, boiled in a water bath for 5 min, and stored at −40 °C until further analysis. Protein aliquots (~20 µg) were loaded in 12% Bis-Tris SurePAGE gels (GenScript Inc., Piscataway, NJ, USA) and run at 200 V with a PowerPac^TM^ Basic (Bio-Rad Laboratories, Hercules, CA, USA) for 35 min at room temperature with Tris-MOPS-SDS running buffer (GenScript, Inc., Piscataway, NJ, USA). After electrophoresis, resolved proteins were transferred onto polyvinylidene fluoride Trans-Blot^®^ Turbo Transfer Pack membranes through a Trans-Blot^®^ Turbo transfer system (Bio-Rad Laboratories, Hercules, CA, USA) set at 2.5 V for 7 min at room temperature. Then, membranes were blocked with 5% skimmed dry milk for 1 h at room temperature, washed three times with Tris-buffered saline containing 0.05% Tween^®^ 20 (TBST) for 10 min, and incubated with GAPDH (1:50,000), Bax (1:5000), Bcl-2 (1:2000), and caspase-3 (1:2000) antibodies overnight at 4 °C. Membranes were washed three times (10 min) with 1 h at room temperature. All the incubations were performed in an orbital TBST before and after incubation with antirabbit or antimouse secondary antibody (1:2000) for shaker. Then, the blots were captured with a C-Digit Blot scanner (Li-Cor Biosciences, Lincoln, NE, USA) using Clarity Max^TM^ Western ECL Substrate solutions (ratio 1:1, *v*/*v*). Image Studio Software (Li-Cor Biosciences, Lincoln, NE, USA) was used for the quantification of the protein band. GAPDH was used as the loading control, and the relative expression was determined as the ratio of the band intensity of each sample to that of the corresponding GAPDH. Data represent the mean and the standard deviation of four biological replicates.

#### 4.9.6. Production of Intracellular Reactive Oxygen Species (ROS)

Intracellular ROS levels were measured by fluorescence spectrophotometry as previously reported in Hong et al. [56], with minor modifications. CRC cells treated as detailed above were washed twice with 100 µL of ice-cold PBS and incubated for 30 min at 37 °C and 5% CO_2_ in darkness with 150 µL of 10 µM H_2_DCFDA in PBS. Fluorescence intensity was recorded at λ_exc_ = 485 nm and λ_em_ = 528 nm wavelengths on a Synergy H1 microplate reader (BioTek Instruments Inc., Winooski, VT, USA). Results were expressed as percentages concerning untreated cells (C−(−)). Data represent the mean and the standard deviation of eight biological replicates.

### 4.10. Statistical Analyses

Data were expressed as mean ± standard deviation. Statgraphics Centurion XVIII (Statgraphics Technologies, The Plains, VA, USA) was used to perform all the statistical analyses. Statistical differences between mean values were determined using analysis of variance (ANOVA) and post hoc Duncan’s test (*p* ≤ 0.05). Pearson correlation coefficients were performed on centred and standardised data to elucidate the relationships among variables of TSPC and antioxidant assays, as well as among percentage of viable cells and intracellular ROS production.

## 5. Conclusions

In this study, the addition of HYD enhanced the nutritional profile and the biological properties of a standard mousse recipe (MH). Mainly, MH increased the amount of bioactive compounds, such as soluble poly- and OLSC, FA, and TSPC that exhibited good stability to simulated gastrointestinal conditions. Similarly, in vitro digestion and HYD also increased MH antioxidant activity, which was positively correlated with TSPC.

HYD and MH showed indirect anticancer properties in two colon cancer cell lines mediated by an immunostimulatory effect in RAW264.7 and THP-1 macrophages. Firstly, HYD and MH (0.5 mg/mL) before and after in vitro digestion enhanced the production of pro-inflammatory cytokines (TNF-α, IL-6, and IL-1β). Secondly, supernatant from WB-stimulated macrophages demonstrated anticancer properties through the reduction of viable CRC cells, with the supernatant from RAW264.7 cells treated with 0.5 mg/mL of digested-HYD being the most antiproliferative treatment. Digested WB samples decreased the colony formation of HCT-116 and HT-29 cells, and WB samples enhanced the capability of RAW264.7 and THP-1 cells spent medium to inhibit colony formation on HCT-116 cells. While a lack of inner mitochondrial membrane potential alteration was found, increased Bax:Bcl-2 ratio, as well as pro-apoptotic Bax and caspase-3 expressions, suggested activation of the mitochondrial apoptotic pathway when CRC cells were treated with macrophage supernatants. Lower intracellular ROS levels were determined when colon cancer cells were exposed to WB-stimulated macrophage supernatants, leading to a lower pro-oxidative state in the colon cancer cell population. In addition, intracellular ROS levels were positively correlated with the percentage of viable colon cancer cells when treated with WB-stimulated macrophage supernatants.

Although these results provide new insights into the immunomodulatory properties and valorisation of WB, the potential signalling pathway by which WB-stimulated macrophage supernatants exhibited antiproliferative properties is still unclear. Thus, further research studies should be performed to fully elucidate the potential anticancer of WB.

## Figures and Tables

**Figure 1 ijms-24-07436-f001:**
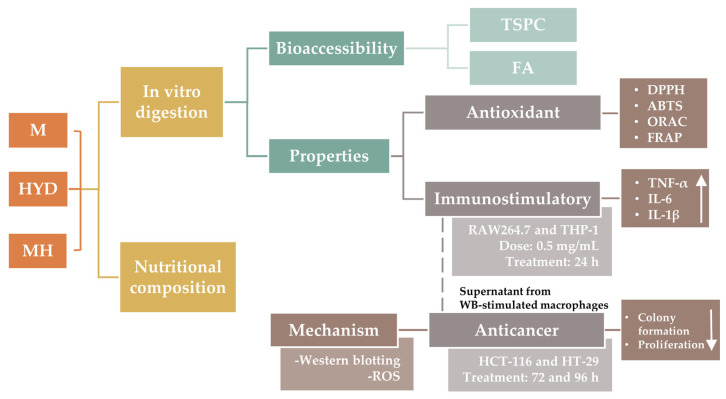
Graphical representation of the experimental design. Abbreviations: control mousse (M); WB hydrolysate (HYD); mousse with WB hydrolysate (MH); total soluble phenolic compounds (TSPC); ferulic acid (FA); 2,2-diphenyl-1-picryl-hydrazyl-hydrate radical scavenging assay (DPPH); 2,2′-azino-bis (3-ethylbenzothiazoline-6-sulfonic acid) radical scavenging assay (ABTS); oxygen radical absorbance capacity (ORAC); ferric reducing antioxidant power (FRAP); tumour necrosis factor (TNF)-α; interleukin (IL); wheat bran (WB); and reactive oxygen species (ROS).

**Figure 2 ijms-24-07436-f002:**
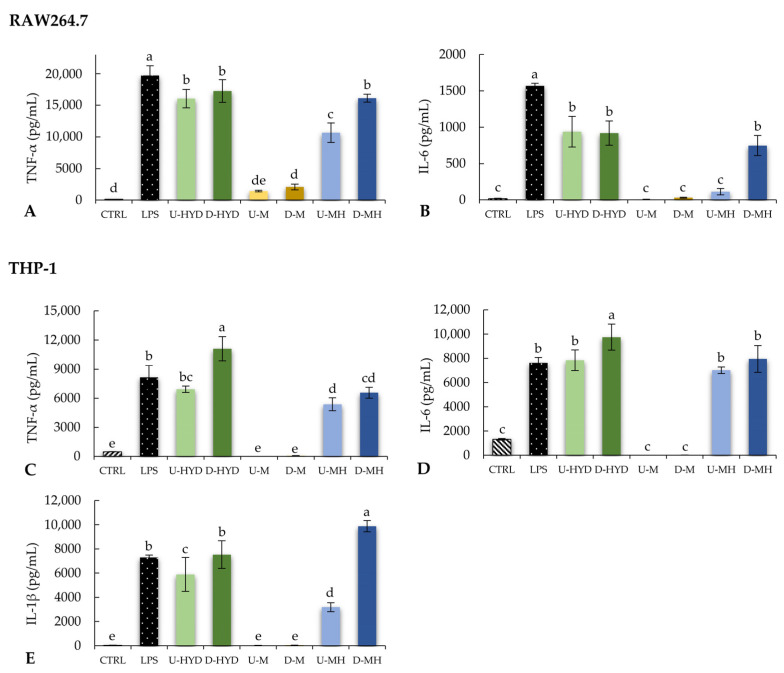
Evaluation of cytokine production by RAW264.7 and THP-1 cells treated with LPS (1 µg/mL) or undigested (U) and digested (D) HYD, M, and MH at 0.5 mg/mL for 24 h. Levels of (**A**) murine tumour necrosis factor (TNF)-α, (**B**) murine interleukin (IL)-6, (**C**) human TNF-α, (**D**) human IL-6, and (**E**) human IL-1β. Bars represent means, and error bars represent standard deviations (n = 8). Different lowercase letters show significant differences from each other (one-way ANOVA, post hoc Duncan’s test, *p* ≤ 0.05). Abbreviations: untreated cells (CTRL); lipopolysaccharide from *E. coli* O55:B5 (LPS); undigested WB hydrolysate (U-HYD); digested WB hydrolysate (D-HYD); undigested control mousse (U-M); digested control mousse (D-M); undigested mousse with WB hydrolysate (U-MH); and undigested mousse with WB hydrolysate (U-MH).

**Figure 3 ijms-24-07436-f003:**
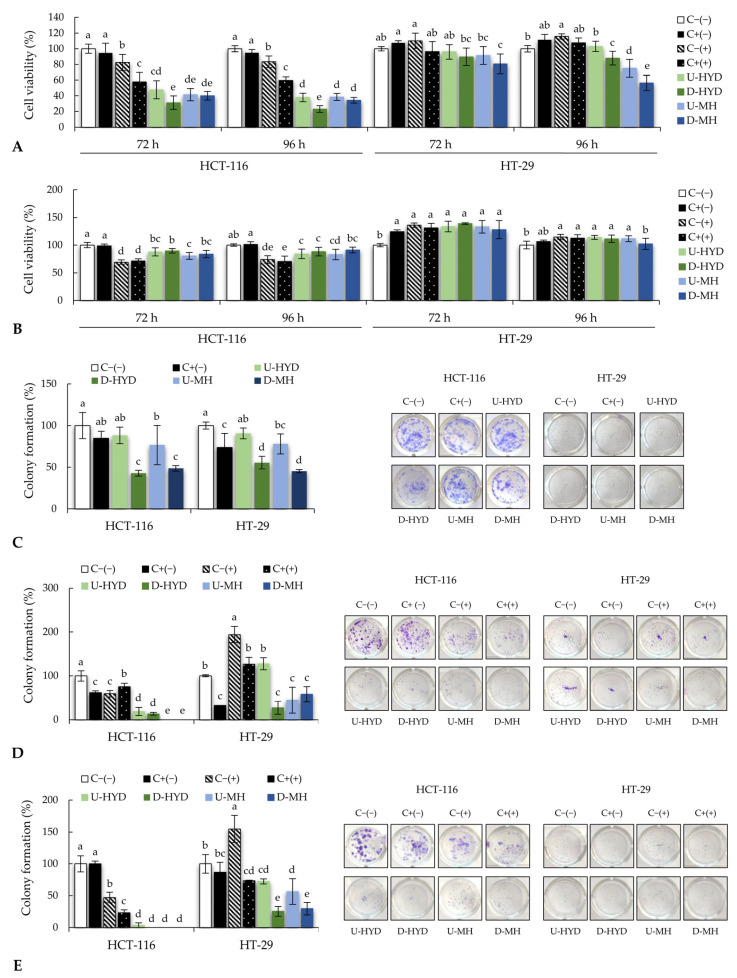
Antiproliferative effect of spent media from (**A**) RAW264.7 and (**B**) THP-1 cells exposed to undigested (U) or digested (D) HYD and MH at 0.5 mg/mL in HCT-116 and HT-29 cells for 72 and 96 h (n = 8). Relative percentage of HCT-116 and HT-29 colonies treated with (**C**) WB samples at 0.5 mg/mL, spent media from (**D**) RAW264.7, and (**E**) THP-1 cells stimulated with WB samples at 0.5 mg/mL (n = 4). **Right side of** (**C**,**D**): representative images from respective human colon cancer cells and treatments. Mean values represented as bars, and standard deviations represented as error bars. Different lowercase letters show significant differences among experimental groups (one-way ANOVA, post hoc Duncan’s test, *p* ≤ 0.05). Abbreviations: colon cancer cells treated with growth media without LPS (C−(−)); with growth media with LPS (1 µg/mL) (C+(−)); with non-stimulated macrophage supernatant (C−(+)); with LPS-stimulated macrophage supernatant (C+(+)); undigested WB hydrolysate (U-HYD); digested WB hydrolysate (D-HYD); undigested mousse with WB hydrolysate (U-MH); and undigested mousse with WB hydrolysate (U-MH).

**Figure 4 ijms-24-07436-f004:**
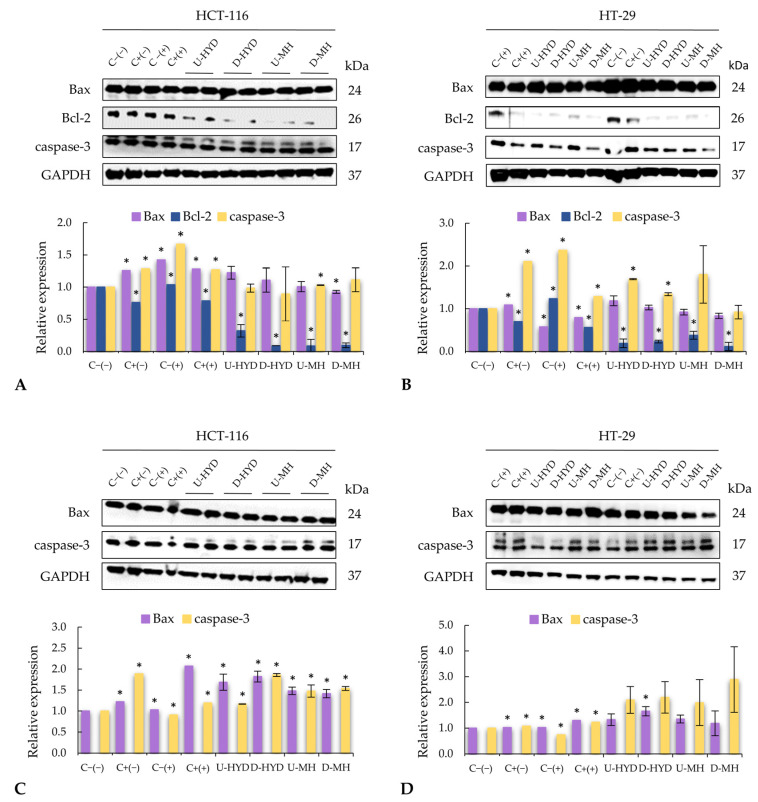
Western blot images and protein relative expression levels of apoptotic proteins in (**A**) HCT-116 and (**B**) HT-29 cells treated with RAW264.7 cells supernatant and western blot images and protein relative expression levels of pro-apoptotic proteins in (**C**) HCT-116 cells and (**D**) HT-29 cells treated with THP-1 supernatant. Each treatment was compared against negative control (C−(−)). Bars represent means, and error bars represent standard deviations (n = 4). * denotes significant differences (one-way ANOVA, post hoc Duncan’s test, *p* ≤ 0.05). Abbreviations: colon cancer cells treated with growth media without LPS (C−(−)); with growth media with LPS (1 µg/mL) (C+(−)); with non-stimulated macrophage supernatant (C−(+)); with LPS-stimulated macrophage supernatant (C+(+)); undigested WB hydrolysate, (U-HYD); digested WB hydrolysate (D-HYD); undigested mousse with WB hydrolysate (U-MH); and undigested mousse with WB hydrolysate (U-MH). Glyceraldehyde 3-phosphate dehydrogenase (GAPDH) was used as housekeeping protein.

**Figure 5 ijms-24-07436-f005:**
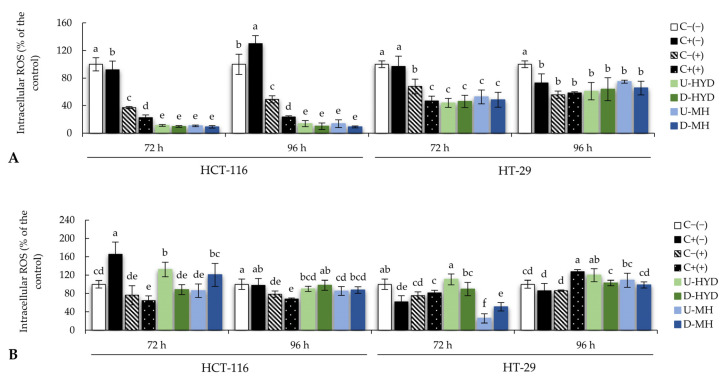
Intracellular ROS production (%) in HCT-116 and HT-29 cells treated with spent media from (**A**) RAW264.7 and (**B**) THP-1 cells exposed HYD and MH before and after digestion at 0.5 mg/mL for 72 and 96 h. Bars represent means, and error bars represent standard deviations (n = 8). Different lowercase letters show significant differences from each other (one-way ANOVA, post hoc Duncan’s test, *p* ≤ 0.05). Abbreviations: colon cancer cells treated with growth media without LPS (C−(−)); with growth media with LPS (1 µg/mL) (C+(−)); with non-stimulated macrophage supernatant (C−(+)); with LPS-stimulated macrophage supernatant (C+(+)); undigested WB hydrolysate (U-HYD); digested WB hydrolysate (D-HYD); undigested mousse with WB hydrolysate (U-MH); and undigested mousse with WB hydrolysate (U-MH).

**Table 1 ijms-24-07436-t001:** Nutritional composition (g/100 g dw) of control mousse (M) and mousse with HYD (MH).

Components	M	MH
Carbohydrates	83.50 ± 2.01	87.50 ± 0.40 *
TDF	9.56 ± 0.04	23.72 ± 0.89 *
IDF	8.27 ± 0.03	1.95 ± 0.34 *
HMW-SDF	nd	1.68 ± 0.26
Oligosaccharides	1.29 ± 0.01	20.10 ± 0.82 *
Protein	1.93 ± 0.02	1.88 ± 0.02 *
Fat	0.27 ± 0.03	0.63 ± 0.03 *
Ash	0.92 ± 0.15	1.04 ± 0.07
Phytic acid	0.22 ± 0.01	0.69 ± 0.04 *

Data are mean values ± standard deviation of three replicates. * denotes statistical differences between means of M and MH (one-way ANOVA, post hoc Duncan’s test, *p* ≤ 0.05). Abbreviations: control mousse (M); mousse with HYD (MH); total dietary fibre (TDF); insoluble dietary fibre (IDF); and high-molecular-weight soluble dietary fibre (HMW-SDF); not detected (nd).

**Table 2 ijms-24-07436-t002:** TSPC and FA (free form) in HYD and mousses (M and MH) before (U) and after (D) simulated gastrointestinal digestion.

Per 100 g dw	U-HYD	D-HYD	U-M	D-M	U-MH	D-MH
TSPC (mg GAE)	783.28 ± 17.55	1396.75 ± 119.11 *	131.10 ± 10.31	99.50 ± 4.50 *	179.32 ± 13.23	144.90 ± 15.16 *
*trans*-FA (mg)	290.56 ± 17.49	248.16 ± 15.28 *	nd	nd	8.87 ± 0.99	9.24 ± 0.48
*cis*-FA (mg)	2.08 ± 0.30	1.99 ± 0.11	nd	nd	0.15 ± 0.02	0.19 ± 0.01
Total FA (mg)	292.64 ± 17.49	250.15 ± 15.38 *	nd	nd	9.02 ± 1.00	9.42 ± 0.33

Data are mean values ± standard deviation of three replicates. * denotes statistical differences between means of undigested (U) and digested (D) samples (one-way ANOVA, post hoc Duncan’s test, *p* ≤ 0.05). Abbreviations: undigested WB hydrolysate (U-HYD); digested WB hydrolysate (D-HYD); undigested control mousse (U-M); digested control mousse (D-M); undigested mousse with WB hydrolysate (U-MH); undigested mousse with WB hydrolysate (U-MH); ferulic acid (FA); gallic acid equivalents (GAE); not detected (nd); and total soluble phenolic compounds (TSPC); not detected (nd).

**Table 3 ijms-24-07436-t003:** Antioxidant activity in HYD and mousses (M and MH) before (U) and after (D) simulated gastrointestinal digestion.

per 100 g dw	U-HYD	D-HYD	U-M	D-M	U-MH	D-MH
DPPH (mmol TE)	5.48 ± 0.25 ^b^	11.79 ± 0.82 ^a^	0.65 ± 0.01 ^f^	2.60 ± 0.05 ^d^	1.33 ± 0.13 ^e^	4.12 ± 0.35 ^c^
ABTS^•+^ (mmol TE)	10.65 ± 0.43 ^d^	20.34 ± 0.77 ^c^	1.82 ± 0.17 ^f^	23.56 ± 1.16 ^b^	6.75 ± 0.19 ^e^	27.32 ± 1.98 ^a^
ORAC (mmol TE)	22.30 ± 2.76 ^b^	30.84 ± 1.70 ^a^	2.80 ± 0.16 ^f^	9.06 ± 1.40 ^d^	6.39 ± 0.20 ^e^	14.94 ± 1.67 ^c^
FRAP (mmol Fe^2+^)	3.07 ± 0.15 ^c^	4.02 ± 0.18 ^b^	0.95 ± 0.02 ^d^	3.02 ± 0.13 ^c^	3.41 ± 0.16 ^c^	5.18 ± 0.85 ^a^

Data are mean values ± standard deviation of three replicates. Different letters in the same row indicate significant differences (one-way ANOVA, post hoc Duncan’s test, *p* ≤ 0.05). Abbreviations: undigested WB hydrolysate (U-HYD); digested WB hydrolysate (D-HYD); undigested control mousse (U-M); digested control mousse (D-M); undigested mousse with WB hydrolysate (U-MH); undigested mousse with WB hydrolysate (U-MH); 2,2-diphenyl-1-picryl-hydrazyl-hydrate radical scavenging assay (DPPH); 2,2′-azino-bis (3-ethylbenzothiazoline-6-sulfonic acid) radical scavenging assay (ABTS); oxygen radical absorbance capacity (ORAC); ferric reducing antioxidant power (FRAP); and 6-hydroxy-2,5,7,8-tetramethyl-2-carboxylic acid (Trolox) equivalents (TE).

## Data Availability

The data used to support the findings of this study are included within the article.

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
