# Peer review of "Antioxidant, Immunostimulatory, and Anticancer Properties of Hydrolyzed Wheat Bran Mediated through Macrophages Stimulation"

_ijms, 2023, doi:10.3390/ijms24087436_

Round 1

Reviewer 1 Report

The manuscript "Anticancer Activity of Hydrolyzed Wheat Bran Mediated Through Macrophages Stimulation" aimed to evaluate the anticancer effect of wheat bran hydrolysate and an enriched-product before and after in vitro digestion in different cellular models.

In my opinion the manuscript is very interesting and original, it is clear and accurate and well-written. It is complete in all its section and I have no specific scientific comments. It has been a pleasure to revise this manuscript

It could be useful:

- provide a graphical representation of the experimental design, in order to faciltate the comprehension by the readers;

- explain why the dose of 0.5 mg/mL has been chosen for cell treatments;

- check the title of chapter 2.5.2;

- check the abbreviation used in the manuscript: they have to be defined the first time they appear in the text

Reviewer 2 Report

lines 69-70:  "WB contains high-value health-promoting components

including dietary fibre and phenolic compounds some of them [9]."

-->  WB contains high-value health-promoting components including

dietary fibre and phenolic compounds.

page 3:  TDF and IDF are used in line 106, but they are not defined until

the legend to Table 1.

Legends to Figures 1-4:  Include the number of samples represented by 

the bars and define the error bars (SD, SEM, range ?).

Legend to Figure 2, line 248:  "Right side of each plot" --> Right side of C-D.

line 372:  "valorisation" --> conversion

line 395:  "by" --> of

line 395:  "3000 rpm"  Convert to x g.

pages 17-19:  Once the source of an instrument has been identified, it 

need not be repeated.  (eg. lines 658-659: Spectrostar Omega: BMG

Labtech, Ortenberg, Germany) is repeated in lines 668-669.

6, line 629: "3000 rpm" Convert this to x

g.

page 17 -19: Once the source of an instrument is identified (eg, lines 658-9.

Author Response

Comments and Suggestions for Authors

  • lines 69-70:  "WB contains high-value health-promoting components including dietary fibre and phenolic compounds some of them [9]." -->  WB contains high-value health-promoting components including dietary fibre and phenolic compounds.

Authors’ response: This sentence has been modified (lines 73-74).

  • page 3:  TDF and IDF are used in line 106, but they are not defined until the legend to Table 1.

Authors’ response: Total dietary fibre (TDF) and insoluble dietary fibre (IDF) have been defined (lines 121-122).

  • Legends to Figures 1-4:  Include the number of samples represented by the bars and define the error bars (SD, SEM, range ?).

Authors’ response: The meaning of bars and error bars has been included (lines 200, 260, 336 and 377) and the number of samples (n) has been included (lines 200, 260, 336 and 377)).

  • Legend to Figure 2, line 248:  "Right side of each plot" --> Right side of C-D.

Authors’ response: This has been addressed (line 259).

  • line 372:  "valorisation" --> conversion

Authors’ response: This has been addressed (line 386).

  • line 395:  "by" --> of

Authors’ response: This has been addressed (line 410).

  • line 395:  "3000 rpm"  Convert to x g.

Authors’ response: This has been addressed (line 637).

  • pages 17-19:  Once the source of an instrument has been identified, it need not be repeated.  (eg. lines 658-659: Spectrostar Omega: BMG Labtech, Ortenberg, Germany) is repeated in lines 668-669.

Authors’ response: This has been addressed (lines 655, 667, 675, 709, 730, 753).

Reviewer 3 Report

V. P. Dia and Co-workers submitted a manuscript about the anticancer activity of hydrolyzed wheat bran by macrophages stimulation. I consider the paper suitable for its publication in the Int. J. Mol. Sci. practically in its present form because it has many interesting features that support my opinion, for instance, I found enough novelty and originality coming from this work, introduction shows suitably the current context of the work, references cited are enough and pertinent, grammar and style are fine, results and discussion sections are well discussed, materials and methods sections are described in detail, statistical treatment of data seems to be Okay, conclusions are well supported by results, and finally, Authors have sufficient experience on the topic, guaranteeing a well performed and conducted research. Thus, the next minor points could be attended before final editing.

- I respectfully suggest modifying a little the title by ‘Antioxidant, Immunostimulatory and Anticancer Properties of Hydrolyzed Wheat Bran Mediated Through Macrophages Stimulation’ because it represents more and integrates better all the work done

- The sharpness of all plots from figures 1 to 4 must be substantially increased (TIFF over 1200 dpi)

- The conclusion section should not start with 'in summary'

Author Response

Comments and Suggestions for Authors

  1. P. Dia and Co-workers submitted a manuscript about the anticancer activity of hydrolyzed wheat bran by macrophages stimulation. I consider the paper suitable for its publication in the Int. J. Mol. Sci. practically in its present form because it has many interesting features that support my opinion, for instance, I found enough novelty and originality coming from this work, introduction shows suitably the current context of the work, references cited are enough and pertinent, grammar and style are fine, results and discussion sections are well discussed, materials and methods sections are described in detail, statistical treatment of data seems to be Okay, conclusions are well supported by results, and finally, Authors have sufficient experience on the topic, guaranteeing a well performed and conducted research. Thus, the next minor points could be attended before final editing.
  • I respectfully suggest modifying a little the title by ‘Antioxidant, Immunostimulatory and Anticancer Properties of Hydrolyzed Wheat Bran Mediated Through Macrophages Stimulation’ because it represents more and integrates better all the work done

Authors’ response:  This has been addressed (line 1).

  • The sharpness of all plots from figures 1 to 4 must be substantially increased (TIFF over 1200 dpi)

Authors’ response: The sharpness of all plots from figures 1-4 has been increased.

  • The conclusion section should not start with 'in summary'

Authors’ response: “In summary” has been replaced by “In this study” (line 802).